# Genetic Characteristics of Avian Influenza Virus Isolated from Wild Birds in South Korea, 2019–2020

**DOI:** 10.3390/v13030381

**Published:** 2021-02-27

**Authors:** Eun-Jee Na, Young-Sik Kim, Sook-Young Lee, Yoon-Ji Kim, Jun-Soo Park, Jae-Ku Oem

**Affiliations:** Laboratory of Veterinary Infectious Disease, College of Veterinary of Medicine, Jeonbuk National University, Iksan 54596, Korea; ejna1212@naver.com (E.-J.N.); yoksik@naver.com (Y.-S.K.); sylee163@jbnu.ac.kr (S.-Y.L.); kimyoonji102@naver.com (Y.-J.K.); spinyang@naver.com (J.-S.P.)

**Keywords:** avian influenza virus, wild bird, surveillance, reassortment

## Abstract

Wild aquatic birds, a natural reservoir of avian influenza viruses (AIVs), transmit AIVs to poultry farms, causing huge economic losses. Therefore, the prevalence and genetic characteristics of AIVs isolated from wild birds in South Korea from October 2019 to March 2020 were investigated and analyzed. Fresh avian fecal samples (3256) were collected by active monitoring of 11 wild bird habitats. Twenty-eight AIVs were isolated. Seven HA and eight NA subtypes were identified. All AIV hosts were *Anseriformes* species. The HA cleavage site of 20 representative AIVs was encoded by non-multi-basic amino acid sequences. Phylogenetic analysis of the eight segment genes of the AIVs showed that most genes clustered within the Eurasian lineage. However, the HA gene of H10 viruses and NS gene of four viruses clustered within the American lineage, indicating intercontinental reassortment of AIVs. Representative viruses likely to infect mammals were selected and evaluated for pathogenicity in mice. JB21-58 (H5N3), JB42-93 (H9N2), and JB32-81 (H11N2) were isolated from the lungs, but JB31-69 (H11N9) was not isolated from the lungs until the end of the experiment at 14 dpi. None of infected mice showed clinical sign and histopathological change in the lung. In addition, viral antigens were not detected in lungs of all mice at 14 dpi. These data suggest that LPAIVs derived from wild birds are unlikely to be transmitted to mammals. However, because LPAIVs can reportedly infect mammals, including humans, continuous surveillance and monitoring of AIVs are necessary, despite their low pathogenicity.

## 1. Introduction

Avian influenza viruses (AIVs), belonging to the *Orthomyxoviridae* family, are enveloped viruses containing an eight-segment, negative-sense RNA genome. AIVs are classified into 16 hemagglutinin (HA) and nine neuraminidase (NA) subtypes based on the presence of these two surface glycoproteins [1]. AIVs have been divided into low-pathogenic (LPAIVs) and high-pathogenic (HPAIVs) groups on the basis of molecular characteristics and pathogenicity in chickens. In particular, the H5 and H7 subtypes of LPAIVs can mutate and evolve into HPAIVs [2]. In South Korea, the first outbreaks of LPAIVs and HPAIVs occurred in 1996 and 2003, respectively [3,4]. These outbreaks frequently occur in poultry farms and cause enormous economic damage.

Wild birds, such as *Anseriformes* and *Charadriiformes*, have been shown to be natural reservoirs of AIVs [5]. Some wild ducks exhibit mild or no clinical symptoms when with HPAIVs, but excrete viruses at higher titers than do chickens [6]. Previous studies have shown that wild birds transmit AIVs to domestic poultry [7,8].

The East Asian-Australian flyway, which supports the greatest populations and diversity of migratory birds [9], includes territories of 22 countries. South Korea is located within this flyway [10]. There is a high possibility that AIVs in domestic chickens were introduced to South Korea through wild birds. Therefore, in this study, we analyzed the prevalence and genetic characteristics of AIVs isolated from wild birds in 2019 and 2020.

AIVs often infect mammals, including humans [11]. Point mutations and segment reassortment of AIVs generate novel strains that can adapt to new hosts. AIV genes have been shown to be incorporated into human influenza viruses responsible for pandemics in the last century [12]. Therefore, in this study, we selected representative viruses to evaluate for pathogenicity in mice because it has been reported that mammals, including humans, can be infected with AIVs despite their low pathogenicity.

## 2. Materials and Methods

### 2.1. Sample Collection and Virus Isolation

Fresh avian fecal samples (3256) were collected between October 2019 and March 2020 by active monitoring of 11 wild bird habitats: JoenJucheon, Sapgyocheon, Gokgyocheon, Bonggangcheon, Bunamho, Ganwolho, Pungseocheon, Geumgang, Cheongmicheon, Mangyeonggang, and Dongjingang. Sample collection sites included Jeolla-do and Chungcheong-do in western South Korea. The locations of the sampling sites are shown in Figure 1. To prevent fecal samples from being contaminated by multiple species, we collected well-distinguished single fecal samples using wooden chopsticks per each fecal sample. Once the wooden chopsticks was used, it was discarded. Samples were immediately transported to the laboratory and stored at 4 °C until further analysis.

### 2.2. Virus Isolation and Identification

Each sample was suspended in phosphate-buffered saline (pH 7.2) containing antibiotics (100 U/μL of penicillin and 100 U/μL of streptomycin) and centrifuged at 3000 rpm for 10 min. The 0.45-μm filtered supernatants were inoculated into 9–11-day-old specific pathogen-free (SPF) embryonated chicken eggs (Seng-Min Inc., Hwaseong, Korea) and incubated at 37 °C for 72 h. Allantoic cavity fluids were harvested for hemagglutinin assays. To detect the virus, a quantitative real-time reverse transcription polymerase chain reaction (qRT-PCR) kit (iNtRON Biotechnology, Seongnam, Korea) was used to amplify the matrix gene. Viral RNA in positive allantoic fluid was extracted using the Viral DNA/RNA Mini Kit (Wizbiosolutions, Seongnam, Korea). To determine subtypes, qRT-PCR was performed using the TOPreal™ One-step RT qPCR Kit (Enzynomics, Daejeon, Korea) with influenza-specific primers and probes [13].

### 2.3. Species Identification

DNA barcoding utilizing the cytochrome C oxidase I (COI) mitochondrial gene was used. Mitochondrial DNA was extracted from fecal samples and specific primers were used to identify bird species [14]: forward (5′-GCATGAGCAGGAATAGTTGG-3′) and reverse (5′-AAGATGTAGACTTCTGGGTG). The PCR conditions were as follows: initial denaturation at 95 °C for 5 min, followed by 35 sequential cycles of 95 °C for 30 s, 56 °C for 1 min, 72 °C for 30 s, and 7 min incubation at 72 °C. The PCR products were sequenced and identified using a BLAST search in GenBank.

### 2.4. Genetic Analysis of AIV Isolates

Genome sequences for H4 and H11 subtypes were obtained through Sanger sequencing. Remaining subtypes were obtained through NGS performed by GnCBIO (Daejeon, Korea) using Illumina HiseqX. RNA of the H4 and H11 subtypes was transcribed into cDNA using a cDNA Synthesis kit (Wizbiosolutions, Seongnam, Korea) according to the manufacturer’s instructions. Synthesized cDNA was amplified by polymerase chain reaction (PCR) using primers for the eight viral genomic segments [15,16] (Table 1). Amplicons were purified using the LaboPass™ Gel Extraction Kit (Cosmo Genetech, Daejeon, Korea) and sequenced at Cosmo Genetech (Daejeon, Korea) with an ABI 3730xl DNA analyzer (Applied Biosystems, Foster City, CA, USA). Nucleotide sequences were translated and aligned using the Clustal W algorithm in BioEdit v.7.0.9.0. Phylogenetic trees were constructed by the neighbor-joining method using MEGA 7 software with 1000 bootstrap replications. Reference sequences were downloaded from the NCBI Influenza Virus Resource (https://www.ncbi.nlm.nih.gov/genbank/, accessed on 24 January 2021) and GISAID (http://www.gisaid.org, accessed on 24 January 2021).

### 2.5. Pathogenicity in Mice

All experiments with mice were conducted in an enhanced animal biosafety level 2+ (ABSL+2) facility. Twelve female six-week-old seronegative BALB/c mice (Samtaco, Osan, Korea) were anesthetized with avertin (Sigma-Aldrich, Milan, Italy) and inoculated intranasally with representative A/mallard/South Korea/JB21-58/2019 (JB21-58), A/Falcated duck/South Korea/JB42-93/2020 (JB42-93), A/spot-billed duck/South Korea/JB32-81/2019 (JB32-81), and A/mallard/South Korea/JB31-69/2019 (JB31-69) viruses in a volume of 50 μL. The inoculum titer was 10^6.5^ EID_50_/mL for JB21-58 (H5N3), JB32-81 (H11N2), and JB31-69 (H11N9), and 10^5.0^ EID_50_/mL for JB42-93 (H9N2). When the JB42-93 was isolated from fecal sample, titer (EID_50_/mL) was lower than JB21-58, JB42-93 and JB32-81. According to previous studies AIVs may get antigenic mutations during egg and cell passage cultivation [17,18]. For this reason, we used JB42-93 despite it was not same inoculum titer with other viruses. Twelve other BALB/c mice were inoculated intranasally with 50 μL PBS as a negative control group. Mice were monitored daily for changes in body weight, body temperature, and other clinical signs. Three mice from each group were euthanized on days 1, 3, 5, and 14 post-infection (dpi); the lungs were collected for titration using an egg-infection dose assay. Lungs were homogenized in 1 mL of cold PBS and clarified at 1300 rpm for 5 min. Next, 150 μL of the supernatant was used to detect virus using a qRT-PCR kit for the matrix gene (iNtRON Biotechnology, Seongnam, Korea). Subsequently, the supernatants were used to determine the egg infectivity dose (EID_50_).

Briefly, 0.2 mL of serially diluted (10^0^–10^4^) supernatants were inoculated into 9–11-day-old specific pathogen-free (SPF) embryonated chicken eggs 72 h after the HA test. The EID_50_ was calculated by the Reed and Muench method. Serum samples were collected at 14 dpi, then seroconversion was assayed using an NP-ELISA kit (Bionote, Hwaseong, Korea). The NP-ELISA kit based on a competitive ELISA method and that OD positive sample become smaller than OD negative control. If the percent inhibition (PI) values were 50 or higher, the sample was considered positive (PI value = [1 − (OD test sample/OD negative control)] × 100). The remaining lung tissues were fixed in 10% formaldehyde solution, dehydrated, embedded in paraffin, sectioned at 3 μm, and processed for hematoxylin and eosin (H&E) staining.

## 3. Results

### 3.1. Virus Isolation

From 3256 wild-bird fecal samples, 28 AIVs were isolated. The 28 isolates are listed in Table 2.

### 3.2. Prevalence of AIVs

The prevalence of avian influenza viruses from October 2019 and March 2020 was 0.85%. Influenza virus subtypes H3-6, H9-11, N1-4, and N6-9 were isolated. Of the 28 isolates, six were mixed subtypes (H3/10N3/8, H3/10N2/7, H3/10NN8, H3N2/7, H5/6N1/3, and H10N2/7). The most frequently detected HA subtype was H4 (25%; n = 7), followed by H11 (18%; n = 5), H6 (11%; n = 3), H10 (11%; n = 3), H5 (7%; n = 2), and H9 (7%; n = 2) except mixed subtypes. The most frequently detected NA subtype was N2 (21%; n = 6), followed by N6 (18%, n = 5), N3 (11%, n = 3), N9 (11%, n = 3), N4 (7%; n = 2), N1 (4%; n = 1), N7 (4%; n = 1), and N8 (4%; n = 1), except mixed subtypes (Table 3). The most frequently detected HA and NA combination was H4N6, which accounted for 17.8% of all isolated AIVs.

The hosts of all AIVs were wild birds belonging to the order *Anseriformes*. The most frequently detected host was *Anas platyrhynchos* (57%; n = 16), followed by *Anas poecilorhyncha* (25%; n = 7), *Anas falcate* (7%, n = 2), *Anas crecca* (4%, n = 1), *Anser albifrons* (4%, n = 1), and *Anser fabalis* (4%, n = 1).

### 3.3. Genetic Characterization of AIV Isolates

Of the 28 isolates, the full or partial sequences of 20 were obtained, except those of mixed subtypes (JB12-8, JB15-42, JB29-40, JB29-56, JB32-41-45, JB36-16-20, JB21-48, and JB43-51-55). The sequences were deposited in the GenBank and the accession number are shown in Appendix A. The amino-acid sequences of the HA cleavage site were PEKASR/GLF for H4 subtypes, PQRETR/GLF for H5 subtypes, PQRETR/GLF for H6 subtypes, PAASNR/GLF for H9 subtypes, EVVQGR/GLF for H10 subtypes, and PAIASR/GLF for H11 subtypes. This observation indicates that all isolates are low-pathogenic AIVs. The residues of the HA gene at positions 190, 225, 226, and 228 (numbering based on H3 A/Aichi/2/1968) were glutamic acid (E), glycine (G), glutamine (Q), and glycine (G), respectively, suggesting more efficient binding to avian receptors than to human receptors [19].

The E627K and D710N substitutions in the PB2 gene, which increased viral polymerase activity in mammalian cell lines, were not detected in any of the isolates [20]. However, three amino-acid substitutions in the PB2 gene, L89V, G309D, and T339K, which increase virulence in mice, were observed in all the isolates [21]. The H436Y and D622G substitutions in the PB1 gene, which increase polymerase activity in mice, were identified in all isolates, but 13 of the 20 had an N66S substitution in the PB1-F2 gene, which enhances virulence in mice [22,23,24] (Table 4).

Additional variants related to increased virulence in mammalian cells and mice, were investigated (Table 4), such as N409S and K615N in the PA gene [25], N319K in the NP gene [26], 54–72 deletions in the NA gene [27], N30D and T215A in the M gene [28], and P42S and D92E in the NS gene [29,30]. The N30D and T215A substitutions in the M gene were identified in all isolates. Substitutions of N409S in the PA gene and P42S in the NS gene were identified in 19 and 13 isolates, respectively. K615N in the PA gene, N319K in the NP gene, 54–72 deletions in the NA gene, and D92E in the NS gene were not identified in any of the isolates.

Residues related to drug resistance were investigated. The I117T substitution in the NA gene was identified in 18 of the 20 isolates, but the R152K and H274Y substitutions in the NA gene were not identified [20,31]. The substitutions of L26F and S31N in the M gene were not identified in any of the isolates [32] (Table 4).

### 3.4. Phylogenetic Analyses

#### 3.4.1. Hemagglutinin

The HA genes of 18 isolates belonged to the Eurasian lineage, but those of the H10 isolates (JB32-15 and JB36-65) belonged to the American lineage (Figure 2). The HA genes of H4, H5, H6, H9, H10, and H11 isolates shared 98.1-99.8%, 99.7%, 83.5%, 99.0%, 99.6%, and 94.2-99.8% nucleotide identities, respectively. The HA genes of H4 and H10 isolates clustered into the same subgroup, despite different NA subtypes. However, the HA genes of H6 and H11 isolates with different NA subtypes showed low similarity and clustered into different subgroups. According to recent study, the HA genes of H5 isolates clustered within the HA-III subgroup, consisting of H5 LPAIVs identified in Korea between 2015-2017 and were genetically distant from H5 HPAIVs [33]. The HA genes of H9 isolates clustered within the Y439-like lineage.

#### 3.4.2. Neuraminidase

The NA genes of all 20 isolates belonged to the Eurasian lineage (Appendix A). The NA genes of the N2, N3, N6, and N9 isolates shared 93.3–99.6%, 99.4–99.8%, 92.8–100%, and 99.8–99.9% nucleotide identities, respectively. Those of the N3 and N9 isolates clustered into the same subgroup, but N2 and N6 clustered into two distinct subgroups. In particular, the N9 genes showed 93.5% nucleotide identity with the N9 gene of A/Anhui/1/2013(H7N9), which caused human infections in China, and clustered within different groups. The N2 genes showed 85.1–86.4% nucleotide identities with the N2 gene of A/HK2108/2008(H9N2), which caused human infections, again clustering within different groups.

#### 3.4.3. Internal Genes

Nearly all alleles of the six internal genes (PB2, PB1, PA, NP, M, and NS) of 20 isolates belonged to the Eurasian lineage, except for the NS-III subgroups, which belonged to the American lineage (Appendix A). The PB2, PB1, PA, NP, M, and NS genes of 20 isolates shared 86.2–100%, 92.8–99.9%, 92.9–100%, 90.9–100%, 94.4–99.8%, and 71.0–100% nucleotide identities, respectively. The PB2 genes clustered into two subgroups and shared nucleotide identities of less than 89.9% between subgroups. The PB1 genes clustered into two subgroups in the phylogenetic tree. The nucleotide identities of the PB1 genes were less than 93.8% between different subgroups. The PA genes clustered into three subgroups with nucleotide identities of less than 97.0% between subgroups. The NP genes clustered into three subgroups with nucleotide identities of less than 95.4% among the different subgroups. The M genes clustered into four subgroups and shared nucleotide identities of less than 97.1% between subgroups. The NS gene has separated into two distinct alleles, A and B [34]. The subgroup NS-I clustered into allele A and subgroups NS-II and NS-III clustered into allele B.

### 3.5. Pathogenicity in Mice

We selected four representative AIVs, JB21-58 (H5N3), JB31-69 (H11N9), JB32-81 (H11N2), and JB42-93(H9N2), from the 20 isolates for inoculation into mice. None of the infected mice showed significant clinical signs, including lack of appetite, weight loss, labored breathing, and hyperthermia (Appendix A). The mice infected with JB31-69, JB32-81, and JB42-93 were all alive at 14 days post-infection (dpi), but one mouse injected with the JB21-58 virus died at 3 dpi. JB21-58, JB32-81, and JB42-93 viruses were isolated from the lungs of infected mice without pre-adaptation (Table 5). The JB31-69 virus could not be isolated from the lungs of any of the inoculated mice. The titer of JB32-81 showed 2.2 log_10_ EID_50_/mL at 1 dpi. The titers of JB21-58 and JB42-93 showed 1.9 log_10_ EID_50_/mL and 2.4 log_10_ EID_50_/mL at 3 dpi, respectively. However, any of viral antigens were not detected in lung of mice at 14 dpi. Mice infected with JB21-58, JB31-69, JB32-81, and JB42-93 did not show any histopathological changes in the lungs at any time point (1, 3, 5, and 14 dpi) (Appendix A) Seroconversion was observed in one of three mice challenged with JB32-81 and JB21-58.

## 4. Discussion

Fine-scale tracking of wild migratory birds has revealed that wild birds from Northern Russia, Mongolia, and Siberia migrate to South Korea for the winter [35]. In particular, most return to the western and mid-central provinces of South Korea [36]. This high density of wild birds is associated with a higher incidence of AIVs in poultry farms. The risk of HPAI occurrence is 3–8-times higher in poultry farms within waterfowl habitats than in poultry farms outside waterfowl habitats [36]. Therefore, it is important to conduct surveillance of AIVs in wild birds to prevent and/or control outbreaks in poultry farms.

To efficiently detect AIVs, surveillance should be conducted in areas where *Anseriformes* and *Charadriiformes* are highly concentrated. As HPAIV is most common in the winter season, it is recommended to conduct surveillance at this time [37,38]. In addition, AIVs replicate in the intestinal tracts of waterfowl and shed high concentrations of influenza viruses in feces [39]. Therefore, we collected feces on riversides, the natural habitat of wild waterfowl, during September 2019 and March 2020 to maximize the detection of AIVs. We isolated 28 LPAIVs, yielding a prevalence of 0.85%. These data are consistent with annual AIV prevalence of 0.2–1.5% for South Korean wild bird habitats from 2010 to 2017 [33].

We found diverse subtypes: seven HA (H3-6, H9-11) and eight NA (N1-4, N6-9). The H1-H7 and H9-H13 subtypes had already been detected in South Korea [40]; however, in this study, H7, H12, and H13 were not isolated. Moreover, we isolated H3, H4, H5, H6, and H9 subtypes, which have been isolated from domestic poultry; therefore, there was a risk of transmission of AIVs from wild birds to poultry farms. The H4 and N2 subtypes were isolated most frequently, similar to data from Europe and North America [41,42].

The hosts of all AIVs were species belonged to the order *Anseriformes*, with *Anas platyrhynchos* (mallard) and *Anas poecilorhyncha* (spot-billed duck) the most common. Likewise, these two species were the most common hosts found in other countries [42,43,44]. Therefore, *Anas platyrhynchos* and *Anas poecilorhyncha* may have a role as sentinel species of AIVs. Screening of these two species is recommended for effective surveillance.

None of the 20 isolates possessed multiple basic residues at the HA cleavage site, indicating that they were LPAIVs. The influenza HA must attach to sialic acid (SA) on the host cell surface to initiate infection. Usually, AIVs do not infect humans because the HA of the human virus binds to α2-6-linked SA, while the AIV HA binds to α2-3-linked SA. However, single-residue substitutions in the HA receptor-binding site (RBS) of AIV change binding specificity, particularly, the E190D, G225D, Q226L, and G228S substitutions [19]. In this study, none of the 20 isolates had any of these substitutions, suggesting that all 20 bind to α2-3-linked SA.

Although the L89V, G309D, and T339K substitutions of the PB2 gene were identified, E627K and D710N, which are the most potent enhancers of polymerase activity in mammalian cells, were not detected. We also investigated several single-residue substitutions associated with increased virulence in mammalian cells and mice. K615N (PA), N319K (NP), 54-72 deletions (NA), and D92E (NS) were not present in any of our 20 isolates. H436Y and D622G (PB1), as well as N30D and T215A (M) were present in all 20 isolates. However, only some isolates had N66S (PB1-F2) (65%), N409S (PA) (95%), and P42S (NS) (65%). In a previous study, variants associated with changes in replication and pathogenicity were confirmed in AIVs of wild birds in Korea [45]. However, further research is needed to determine if the substitutions found in this study are related to time flow.

M2 ion channel blockers and NA inhibitors have been approved for the treatment of influenza viruses. We identified no substitutions associated with antiviral resistance except I117T (NA). Although H274Y (NA) was not detected in this study, this substitution has been disseminated in human influenza viruses worldwide [46]. Therefore, continuous surveillance of AIVs is necessary to identify important variants.

The A/goose/Guangdong/1/1996 (Gs/GD) lineage of HPAIV was first reported in 1996; since then, its descendants have spread to over 80 countries [47]. Phylogenetic analysis revealed that the H5 genes were clearly separated from the Gs/GD-like lineage and were closely related to A/duck/Mongolia/926/2019(H5N3). In Eurasia, the LPAI H9N2 virus can be separated into several genetic groups: G1-like, Y280-like, Korean-lineage, and Y439-lineage [48]. The H9 gene isolated in our previous study belonged to the Y439-lineage, which has endemically circulated in poultry in South Korea [49].

In 2013, a novel avian-origin H7N9 virus emerged in China, causing severe respiratory syndrome in humans [50]. Phylogenetic analysis showed that the N9 genes of the three H11N9 isolates were clearly separated from the N9 genes of H7N9. In 2003, an H9N2 influenza virus was isolated from a child with mild respiratory syndrome; genetic analysis revealed that it was of purely avian origin [51]. However, the N2 genes of the six isolates were clustered within a distinct N2 clade of the H9N2 virus. These results demonstrate that the isolated AIVs have low similarity with influenza viruses that cause human infection.

Most of the eight internal genes of our 20 isolates were clustered in Eurasian lineages. However, the HA gene of H10 and the NS genes of JB19-19, JB13-47, JB13-25, and JB42-113 clustered in the American lineage. In addition, the six internal genes, except HA and NA, were divided into multiple subgroups. These results are consistent with active intercontinental reassortment. Moreover, they indicate that AIVs undergo reassortment in other countries. The introduction of a novel gene by reassortment is likely to contribute to host adaptation.

In 1997, an influenza H5N1 virus suggested derived from AIVs was first detected in human [52]. Since then, 881 human infection cases of H5N1 viruses has been identified by 2019 with 52.4% of fatality risk [53]. 23 cases of human infection of HPAIV H5N6 viruses have been reported from 2014 to 2019 and the mortality rate of infection was high (70%) [54]. 50 cases of H9N2 viruses human infection by 2019 were documented since the first case reported in Guangdong in 1988 [55,56]. Compared to H5 subtype viruses, clinical sign of H9N2 human infection were mild respiratory syndrome. Although, H9 subtype viruses were considered less important than H5 subtype viruses, human infection cases of H9N2 have shown increasing tendency in recent years [56]. H11 subtype AIVs has not been isolated in human, antibody of H11 was found in human who has many opportunities to contact birds [57,58]. These reports suggest that H11 of AIVs may infect direct to human.

Therefore, we selected four AIVs, H5N3 (JB21-58), H9N2 (JB42-93), H11N2 (JB32-81), and H11N9 (JB31-69), which are likely to infect mammals, for murine pathogenicity assays. JB21-58, JB42-93, and JB32-81 were reisolated in the lung, whereas JB31-69 was not reisolated through 14 dpi. We observed seroconversion in one of three mice inoculated with JB32-81 and JB21-58 (Table 5). Seroconversion was not observed in any of the three mice challenged with JB42-93, but the percent inhibition (PI) values of two of the three mice were close to the positive PI value (46.1 and 48.5, respectively). Although JB32-81, JB21-58, and JB42-93 were reisolated from the lungs of inoculated mice, no histopathological lesions were observed. In this study, four AIVs were less likely to infect mammals, since mouse pathogenicity was low. However, according to a recent study, H9N2 AIVs that can be directly transmitted to guinea pigs and ferrets has been detected from wild birds. The viruses even can infect mammalian models via respiratory droplets [59]. Novel AIVs that can be transmitted to humans may emerge in the future.

There were no HPAIV outbreaks in domestic poultry farms in the winters of 2019 and 2020 in South Korea [60]. During this period, HPAIVs were not detected in wild bird feces. These results provide indirect evidence that HPAIVs are introduced into South Korea by migratory wild birds during the winter. Therefore, continuous and active surveillance of AIVs from wild birds is prudent to prevent outbreaks of AIVs in poultry farms and may provide early alarms.

## Figures and Tables

**Figure 1 viruses-13-00381-f001:**
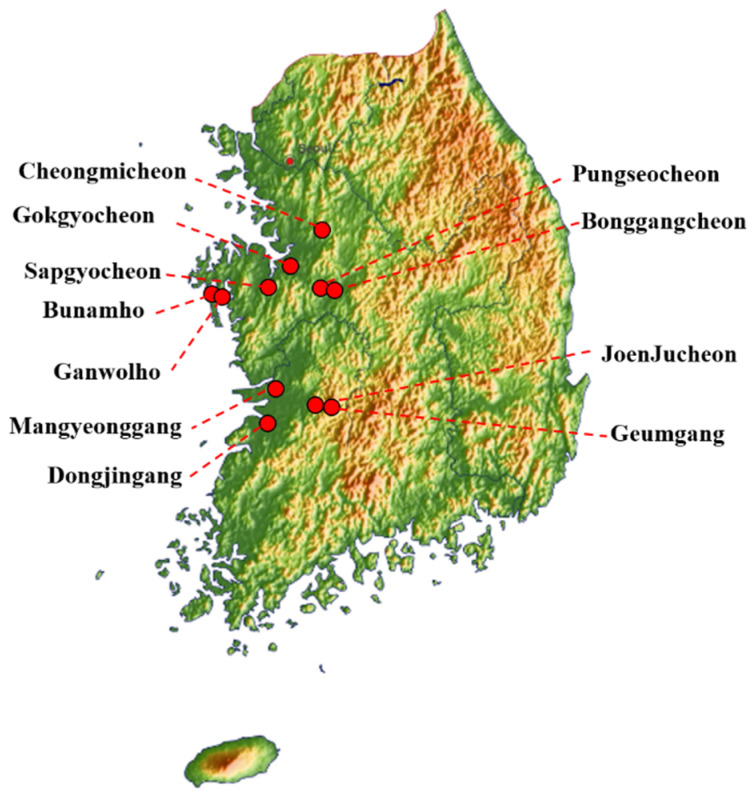
South Korean sampling sites.

**Figure 2 viruses-13-00381-f002:**
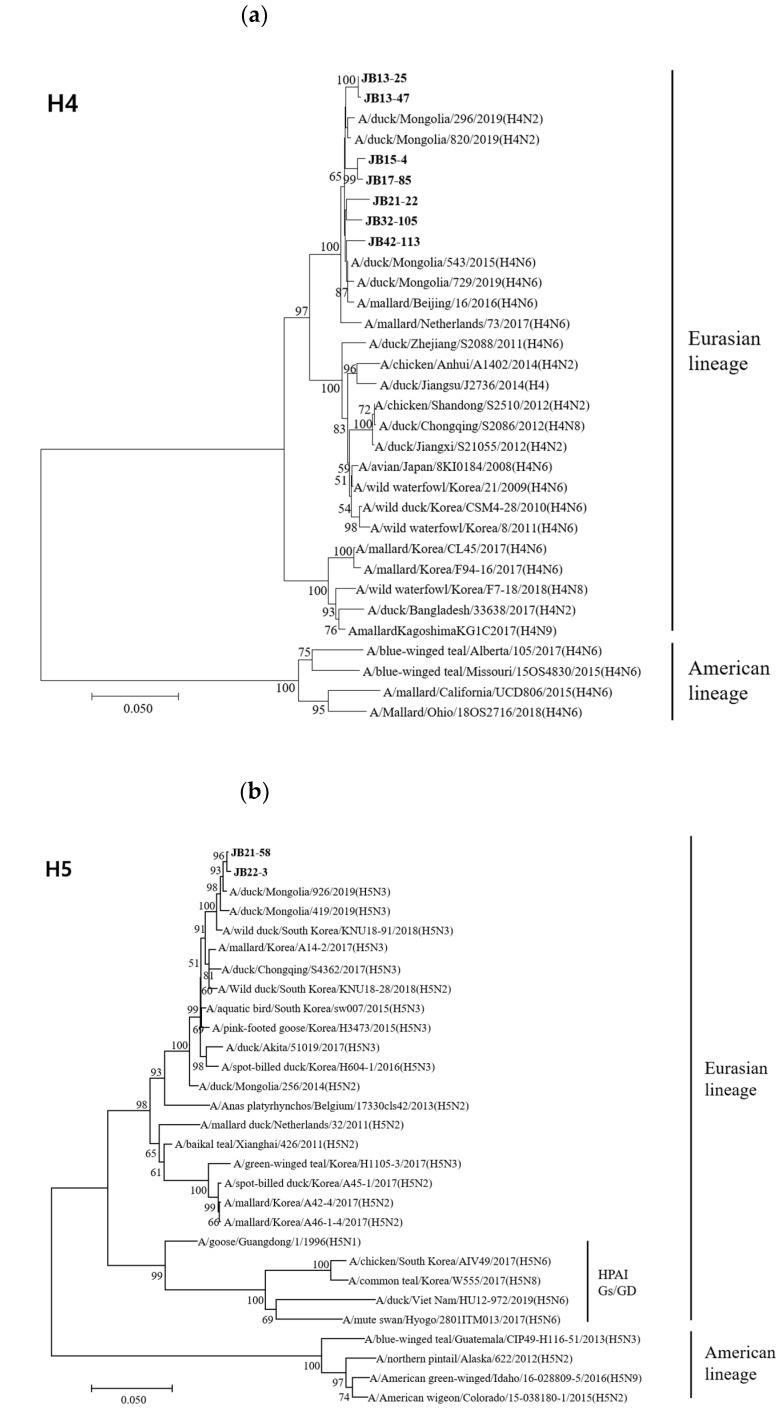
Phylogenetic trees based on the nucleotide sequences of H4 (**a**), H5 (**b**), H6 (**c**), H9 (**d**), H10 (**e**) and H11 (**f**) genes. Trees were produced by the neighbor-joining method and bootstrapped with 1000 replicates. Branch lengths are numbers of nucleotide substitutions per site; only bootstrap values at or above 50% are shown. Isolates identified in this study are shown in bold.

**Table 1 viruses-13-00381-t001:** PCR primers for H4 and H11 avian influenza virus subtypes.

Genes	Primer	Primer Sequences	References
PB2	Bm-PB2-1	5′-TATTGGTCTCAGGGAGCGAAAGCAGGTC-3′	[15]
	PB2-1250R	5′-TCYTCYTGTGARAAYACCAT-3′	[16]
	PB2-1105F	5′-TAYGARGARTTCACAATGGT-3′	
	Bm-PB2-2341R	5′-ATATGGTCTCGTATTAGTAGAAACAAGGTCGTTT-3′	[15]
PB1	Bm-PB1-1	5′-TATTCGTCTCAGGGAGCGAAAGCAGGCA-3′	
	PB1-800R	5′-TCRAGYTTYTCACAKATGCTCC-3′	customized primer
	PB1-560F	5′-ARATACCNGCAGARATGCT-3′	[16]
	PB1-1610R	5′-ACTGTAACHCCAATGCTCAT-3′	customized primer
	PB1-1124F	5′-ARATACCNGCAGARATGCT-3′	[16]
	Bm-PB1-2341R	5′-ATATCGTCTCGTATTAGTAGAAACAAGGTACTT-3′	[15]
PA	Bm-PA-1	5′-TATTCGTCTCAGGGAGCGAAAGCAGGTAC-3′	
	PA-1498R	5′-TNGTYCTRCAYTTGCTTATCAT-3′	[16]
	PA-747F	5′-CATTGAGGGCAAGCTTTC-3′	
	Bm-PA-2233R	5′-ATATCGTCTCGTATTAGTAGAAACAAGGTACTT-3′	[15]
HA	Bm-HA-1	5′-TATTCGTCTCAGGGAGCAAAAGCAGGGG-3′	[15]
	H4-1085R	5′-GGCCTTGCCATCCATTCTCTAT-3′	customized primer
	H4-865F	5′-GGCTCATGTGTCAGTAAG-3′	customized primer
	H11-771F	5′-ACAGGCTGGACGATGAC-3′	customized primer
	H11-881R	5′-CACTT GTAGAGCATGATTC-3′	customized primer
	H11-1100F	5′-YTRATYAATGGWTGGTAYGG-3′	customized primer
	H11-1166R	5′-TGGTCTATCGCCTTCTGA-3′	customized primer
	Bm-NS-890R	5′-ATATCGTCTCGTATTAGTAGAAACAAGGGTGTTTT-3′	[15]
NP	Bm-NP-1	5′-TATTGGTCTCAGGGAGCAAAAGCAGGAGT-3′	
	Bm-NP-1565R	5′-ATATCGTCTCGTATTAGTAGAAACAAGGGTATTTTT-3′	
NA	Ba-NA-1	5′-TATTGGTCTCAGGGAGCAAAAGCAGGAGT-3′	
	N2-848R	5′-TCTCTGCARACACATCTGACGT-3′	customized primer
	N3-689R	5′- CATTCWGACTCYTGAGTTCT -3′	customized primer
	N6-318F	5′- ATGCAATAAGRATAGGDGA -3′	customized primer
	N9-586R	5′- TGCATATTGACATCCTGGAT-3′	customized primer
	Ba-NA-1413R	5′-ATATGGTCTCGTATTAGTAGAAACAAGGAGTTTTTT-3′	[15]
M	Bm-M-1	5′-TATTCGTCTCAGGGAGCAAAAGCAGGGTG-3′	
	Bm-M-1027R	5′-ATATCGTCTCGTATTAGTAGAAACAAGGTAGTTTTT-3′	
NS	Bm-NS-1	5′-TATTCGTCTCAGGGAGCAAAAGCAGGGTG -3′	
	Bm-NS-890R	5′-ATATCGTCTCGTATTAGTAGAAACAAGGGTGTTTT-3′	

**Table 2 viruses-13-00381-t002:** Avian influenza viruses (AIVs) isolated from wild birds in South Korea between 2019 and 2020.

Isolates	Abbreviation	Subtype	Location	Host
A/mallard/South Korea/JB12-8/2019(H3/10N3/8)	JB12-8	H3/10M3/8	Sapgyocheon	*Anas platyrhynchos*
A/spot-billed duck/South Korea/JB13-25/2019(H4N6)	JB13-25	H4N6	Pungseocheon	*Anas poecilorhyncha*
A/ spot-billed duck/ South Korea /JB13-47/2019(H4N6)	JB13-47	H4N6	Pungseocheon	*Anas poecilorhyncha*
A/ spot-billed duck/ South Korea /JB15-4/2019(H4N3)	JB15-4	H4N3	Gokgyocheon	*Anas poecilorhyncha*
A/ mallard/ South Korea/JB15-42/2019(H5/6N1/3)	JB15-42	H5/6N1/3	Gokgyocheon	*Anas platyrhynchos*
A/ mallard/ South Korea/JB17-85/2019(H4N6)	JB17-85	H4N6	Gokgyocheon	*Anas platyrhynchos*
A/mallard/South Korea/JB19-19/2019(H6N8)	JB19-19	H6N8	JoenJucheon	*Anas platyrhynchos*
A/mallard/South Korea/JB21-22/2019(H4N6)	JB21-22	H4N6	Bonggangcheon	*Anas platyrhynchos*
A/spot-billed duck/South Korea /JB21-48/2019(H6N1)	JB21-48	H6N1	Bonggangcheon	*Anas poecilorhyncha*
A/mallard/South Korea/JB21-58/2019(H5N3)	JB21-58	H5N3	Bonggangcheon	*Anas platyrhynchos*
A/mallard/South Korea/JB22-3/2019(H5N3)	JB22-3	H5N3	Cheongmicheon	*Anas platyrhynchos*
A/mallard/South Korea/JB29-40/2019(H3/10N2/7)	JB29-40	H3/10N2/7	Dongjingang	*Anas platyrhynchos*
A/mallard/South Korea/JB29-56/2019(H3N2/7)	JB29-56	H3N2/7	Dongjingang	*Anas platyrhynchos*
A/mallard/South Korea/JB29-91-95/2019(H11N2)	JB29-91-95	H11N2	Dongjingang	*Anas platyrhynchos*
A/mallard/South Korea/JB31-69/2019(H11N9)	JB31-69	H11N9	Mangyeonggang	*Anas platyrhynchos*
A/spot-billed duck/South Korea/JB31-86-90/2019(H6N2)	JB31-86-90	H6N2	Mangyeonggang	*Anas poecilorhyncha*
A/mallard/South Korea/JB31-96/2019(H11N9)	JB31-96	H11N9	Mangyeonggang	*Anas platyrhynchos*
A/mallard/South Korea/JB31-103/2019(H11N9)	JB31-103	H11N9	Mangyeonggang	*Anas platyrhynchos*
A/Eurasian teal/South Korea/JB32-15/2019(H10N7)	JB32-15	H10N7	Dongjingang	*Anas crecca*
A/mallard/South Korea/JB32-41-45/2019(H3/10N8)	JB32-41-45	H3/10N8	Dongjingang	*Anas platyrhynchos*
A/spot-billed duck/South Korea/JB32-81/2019(H11N2)	JB32-81	H11N2	Dongjingang	*Anas poecilorhyncha*
A/spot-billed duck/South Korea/JB32-105/2019(H11N2)	JB32-105	H11N2	Dongjingang	*Anas poecilorhyncha*
A/Greater White-fronted Goose/South Korea/JB36-16-20/2019(H10N2/7)	JB36-16-20	H10N2/7	Geumgang	*Anser albifrons*
A/Bean goose/South Korea/JB36-65/2019(H10N4)	JB36-65	H10N4	Geumgang	*Anser fabalis*
A/Falcated duck/South Korea/JB42-30/2020(H9N2)	JB42-30	H9N2	Mangyeonggang	*Anas falcata*
A/Falcated duck/South Korea/JB42-93/2020(H9N2)	JB42-93	H9N2	Mangyeonggang	*Anas falcata*
A/mallard/South Korea/JB42-113/2020(H4N6)	JB42-113	H4N6	Mangyeonggang	*Anas platyrhynchos*
A/mallard/South Korea/JB43-51-55/2020(H10N4)	JB43-51-55	H10N4	Mangyeonggang	*Anas platyrhynchos*

**Table 3 viruses-13-00381-t003:** Distribution of avian influenza virus hemagglutinin (HA) and neuraminidase (NA) subtypes. Numbers of mixed subtypes were shown in parentheses.

Subtype	N1	N2	N3	N4	N5	N6	N7	N8	N9	Total
H1	-	-	-	-	-	-	-	-	-	0
H2	-	-	-	-	-	-	-	-	-	0
H3	-	(2)	(1)	-	-	-	(2)	(2)	-	(7)
H4	-	1	1	-	-	5	-	-	-	7
H5	(1)	-	2(1)	-	-	-	-	-	-	2(2)
H6	1(1)	1	(1)	-	-	-	-	1	-	3(2)
H7	-	-	-	-	-	-	-	-	-	0
H8	-	-	-	-	-	-	-	-	-	0
H9	-	2	-	-	-	-	-	-	-	2
H10	-	(2)	(1)	2	-	-	1(2)	(2)	-	3(7)
H11	-	2	-	-	-	-	-	-	3	5
H12	-	-	-	-	-	-	-	-	-	0
H13	-	-	-	-	-	-	-	-	-	0
H14	-	-	-	-	-	-	-	-	-	0
H15	-	-	-	-	-	-	-	-	-	0
H16	-	-	-	-	-	-	-	-	-	0
Total	1(2)	6(4)	3(3)	2	0	5	1(4)	1(4)	3	

**Table 4 viruses-13-00381-t004:** Amino-acid sequences and mutations in avian influenza viruses (AIVs) isolated from wild birds in South Korea between 2019 and 2020. The sequences in which amino acid substitutions were identified are shown in yellow. * HA receptor binding site (RBS) related with binding specificity. ^†^ Amino acid mutations related to increased virulence in mammalian cells and mice. ^§^ Amino acid mutations related to drug resistance.

**Name**	**HA**					**PB2**					**PB1**		**PB1-F2**
	**Cleavage Site**	**E190D ***	**G225D ***	**Q226L ***	**G228S ***	**L89V ^†^**	**G309D ^†^**	**T339K ^†^**	**E627K ^†^**	**D701N ^†^**	**H436Y ^†^**	**D622G ^†^**	**N66S ^†^**
JB13-25	PEKASR/G	E	G	Q	G	V	D	K	E	D	Y	G	N
JB13-47	PEKASR/G	E	G	Q	G	V	D	K	E	D	Y	G	N
JB15-4	PEKASR/G	E	G	Q	G	V	D	K	E	D	Y	G	S
JB17-85	PEKASR/G	E	G	Q	G	V	D	K	E	D	Y	G	S
JB19-19	PQIETR/G	E	G	Q	G	V	D	K	E	D	Y	G	N
JB21-22	PEKASR/G	E	G	Q	G	V	D	K	E	D	Y	G	S
JB21-58	PQRETR/G	E	G	Q	G	V	D	K	E	D	Y	G	S
JB22-3	PQRETR/G	E	G	Q	G	V	D	K	E	D	Y	G	S
JB29-91-95	PAIASR/G	E	G	Q	G	V	D	K	E	D	Y	G	S
JB31-69	PAIASR/G	E	G	Q	G	V	D	K	E	D	Y	G	S
JB31-86-90	PQIETR/G	E	G	Q	G	V	D	K	E	D	Y	G	N
JB31-96	PAIASR/G	E	G	Q	G	V	D	K	E	D	Y	G	S
JB31-103	PAIASR/G	E	G	Q	G	V	D	K	E	D	Y	G	S
JB32-15	EVVQGR/G	E	G	Q	G	V	D	K	E	D	Y	G	N
JB32-81	PAIASR/G	E	G	Q	G	V	D	K	E	D	Y	G	S
JB32-105	PEKASR/G	E	G	Q	G	V	D	K	E	D	Y	G	N
JB36-65	EVVQGR/G	E	G	Q	G	V	D	K	E	D	Y	G	S
JB42-30	PAASNR/G	E	G	Q	G	V	D	K	E	D	Y	G	S
JB42-93	PAASNR/G	E	G	Q	G	V	D	K	E	D	Y	G	S
JB42-113	PEKASR/G	E	G	Q	G	V	D	K	E	D	Y	G	N
**Name**	**PA**		**NP**	**NA**				**M1**		**M2**		**NS1**	
	**N409S ^†^**	**K615N ^†^**	**N319K ^†^**	**54-72 deletion ^†^**	**I117T ^§^**	**R152K ^§^**	**H274Y ^§^**	**N30D ^†^**	**T215A ^†^**	**L26F ^§^**	**S31N ^§^**	**P42S ^†^**	**D92E ^†^**
JB13-25	S	K	N	-	T	R	H	D	A	L	S	A	D
JB13-47	S	K	N	-	T	R	H	D	A	L	S	A	D
JB15-4	S	K	N	-	T	R	H	D	A	L	S	S	D
JB17-85	S	K	N	-	T	R	H	D	A	L	S	S	D
JB19-19	S	K	N	-	I	R	H	D	A	L	S	A	D
JB21-22	S	K	N	-	T	R	H	D	A	L	S	S	D
JB21-58	S	K	N	-	T	R	H	D	A	L	S	A	D
JB22-3	S	K	N	-	T	R	H	D	A	L	S	A	D
JB29-91-95	S	K	N	-	T	R	H	D	A	L	S	S	D
JB31-69	S	K	N	-	T	R	H	D	A	L	S	S	D
JB31-86-90	S	K	N	-	T	R	H	D	A	L	S	S	D
JB31-96	S	K	N	-	T	R	H	D	A	L	S	S	D
JB31-103	S	K	N	-	T	R	H	D	A	L	S	S	D
JB32-15	S	K	N	-	T	R	H	D	A	L	S	S	D
JB32-81	S	K	N	-	T	R	H	D	A	L	S	S	D
JB32-105	S	K	N	-	T	R	H	D	A	L	S	S	D
JB36-65	S	K	N	-	I	R	H	D	A	L	S	A	D
JB42-30	S	K	N	-	T	R	H	D	A	L	S	S	D
JB42-93	S	K	N	-	T	R	H	D	A	L	S	S	D
JB42-113	N	K	N	-	T	R	H	D	A	L	S	A	D

**Table 5 viruses-13-00381-t005:** Viral titers in lungs and seroconversion in mice inoculated intranasally with JB21-58 (H5N3), JB31-69 (H11N9), JB32-81 (H11N2), and JB42-93(H11N2) viruses.

Virus	Inoculum Titer (log_10_ EID_50_ per mL)	Lung Titer ^a^ (log_10_ EID_50_ per mL)				Seroconversion ^c^ (PI)
		1 dpi	3 dpi	5 dpi	14 dpi	14 dpi
JB31-69 (H11N9)	6.5	- ^b^ (3/3)	- ^b^ (1/3)	(0/3)	(0/3)	- ^d^
JB32-81(H11N2)	6.5	2.2 (3/3)	<1 (3/3)	<1 (3/3)	(0/3)	1/3 (67)
JB21-58 (H5N3)	6.5	- ^b^ (2/3)	1.9 (3/3)	- ^b^ (3/3)	(0/3)	1/3 (74.1)
JB42-93 (H9N2)	5.0	- ^b^ (2/3)	2.4 (3/3)	- ^b^ (3/3)	(0/3)	0/3

^a^ The viral titers (log_10_ EID_50_/ mL) of the pooled samples with numbers of animals in which the virus was detected by qRT-PCR per number inoculated shown in parentheses. ^b^ Virus not recovered. ^c^ Numbers of animals in which positive serum was detected per number inoculated, with percent inhibition (PI) values shown in parentheses. ^d^ Data was not shown. PI value = [1 − (OD test sample/OD negative control)] × 100 (positive PI value > 50). The ELISA kit is based on a competitive ELISA method and OD negative control become larger than OD positive control.

## Data Availability

Not applicable.

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
