# Peer review of "Genetic Characteristics of Avian Influenza Virus Isolated from Wild Birds in South Korea, 2019–2020"

_viruses, 2021, doi:10.3390/v13030381_

Round 1
Reviewer 1 Report
The authors present data on wild bird influenza A virus (IAV) infections in Korea. While results from IAV monitoring and surveillance efforts are not uncommon or especially novel, this manuscript takes the next step by reporting details on genetic substitutions and virulence factors in mice. While no earth-shattering results were found (no HPIAVs, no novel variants, no virulence in mice), the paper is still an interesting bit of data and well-structured/well-written. A few specific comments are included in the PDF.

Author Response
Thank you for inviting us to submit a revised draft of our manuscript entitled, "Genetic Characteristics of Avian Influenza Virus isolated from Wild Bird in South Korea, 2019-2020" to Viruses. We also appreciate the time and effort you and each of the reviewers have dedicated to providing insightful feedback on ways to strengthen our paper. Thus, it is with great pleasure that we resubmit our article for further consideration. We have incorporated changes that reflect the detailed suggestions you have graciously provided. We also hope that our edits and the responses we provide below satisfactorily address all the issues and concerns you and the reviewers have noted.
Reviewer 2 Report
Na EJ, et al.
“Genetic characteristics of avian influenza virus isolated from wild birds in South Korea, 2019-2020”
In this manuscript, the authors characterized viruses isolated from feces of wild birds, which were collected for surveillance in 2019-2020 season. According to genetic analyses, there were a variety of subtypes and a variety of substitutions that had been reported as association of virulence in mammals. The authors also examined the pathogenicity of representative 4 viruses in mice, demonstrating that none of infected mice showed clinical symptoms although viruses replicated in lungs of mice at low titers. Given that viruses from wild birds are most likely related to outbreak in domestic poultry farms, surveillance of avian influenza viruses from wild birds, like this study, are important especially for early detection of highly pathogenic viruses (H5 and H7 subtypes). There are some points to be addressed.
Specific comments
Section 3.1.
- Because isolates were listed in Table 2, the authors should simply describe “The 28 isolates were listed in Table 2”. This means they do not need to show all strain names in text.
Section 3.5.
- “Phylogenetic in mice” should be “Pathogenicity in mice”.
- The reason why 4 viruses were selected in this analysis was not clear although they described “which are likely to infect mammals” in line 386. If there are not any specific reasons, it is okay. Please describe so.
- The authors should describe the reason why they didn’t use same titers for this analysis.
- Lines 283-284. “The JB31-69 virus could not be isolated from the lungs of any of the inoculated mice”. However, Table 5 showed that viruses were detected at 1 and 3 dpi in lungs of mice inoculated with this virus.
- Line 287. “showed the highest titer in the lung at 3dpi”. Because there are several columns showing “ND” regarding virus titers in lungs, this description is not suitable. To indicate this more precisely, the authors should show average titer based on each titer of individual mouse, not titers of pooled samples because virus titer in lungs of each mouse generally varies, especially in case virus is low pathogenic to mice.
Table 5.
- What does it mean “-”?
- PI value. In case a test sample is generally positive with ELISA, OD test sample must be greater than OD negative control sample. If this is the case, PI value should be negative number when the formula shown in Table 5 is used. Please re-check PI value.
Discussion.
- Line 360. “H274Y (NA) was detected in this study” should be “H274Y (NA) was not detected in this study”.
- Lines 388-389. As described above, the authors should show average titer based on individual titer if they would like to describe this.
- 50. This article has been retracted (https://link.springer.com/article/10.1007/s12250-020-00248-9).
Minor points
- Table 2. First virus; A/mallard/South Korea/JB12-8/2019 (H3/10M3/8). “M” should be “N”.
- Table 3. Please explain parenthesis and * in footnote.
- Table 4. Please explain yellow-highlighted in footnote.
- Line 232. “antigenically” should be “genetically” because the authors have not examined antigenicity.
- Line 259. Because “A/chicken/Hunan/4246/2005” should be isolated from chicken, not human, this virus did not cause human infection. Please modify the sentence.
- Line 398. “HPPAIV” should be “HPAIV”.
Author Response

(The authors gave the same response as above.)

Round 2
Reviewer 2 Report
Na EJ, et al.
“Genetic characteristics of avian influenza virus isolated from wild birds in South Korea, 2019-2020”
The manuscript has been revised well and the authors' responses are clear and satisfying. However, there are some additional comments as follows. I hope these comments will be helpful.
Specific comments
Section 2.5. Regarding NP-ELISA kit.
- This reviewer understood the authors’ explanation. However, many readers may not be able to understand without their explanation. Thus, the authors should describe that this kit is based on a competitive ELISA method and that OD positive sample become smaller than OD negative control. In addition, regarding “percent inhibition (PI) values”, they should explain it with the formular there as well as a footnote of Table 5.
Table 5.
- This reviewer understood how to read this table by the authors’ explanation. However, as it stands, this table misleads the reader. The followings are suggestions:
- Titer should be outside parenthesis, but number should be in parenthesis, because this column indicates “Lung titer”.
- “ND” should be “-” instead of “not done”, and explain like “-“ stands for “virus not recovered” in a footnote.
- “b” should be attached to number and explain like “bNumber of animals in which the virus was detected by qRT-PCR per number inoculated”.
Supplemental Figure S1 (a).
- As this reviewer pointed out, “chicken” in an isolate name means that the virus was isolated from chicken, not human. Thus, vertical line with “Human” in this figure should not cover “chicken” virus even if their nucleotide identities are high.
Author Response
Dear Lewis Everett
Thank you for considering my article for publication in Viruses. I am grateful to you and the reviewers for the valuable suggestion provides.
Here are responses to the reviewer comments:
The revised sentences were showed gray.
Comment 1. Section 2.5. Regarding NP-ELISA kit. This reviewer understood the authors’ explanation. However, many readers may not be able to understand without their explanation. Thus, the authors should describe that this kit is based on a competitive ELISA method and that OD positive sample become smaller than OD negative control. In addition, regarding “percent inhibition (PI) values”, they should explain it with the formular there as well as a footnote of Table 5.
RESPONSE: We have reflected this comment by describing additional explanation of ELISA kit in line 138-142 (section 2.5.). Also, we add additional explanation of ELSIA kit with PI formular in footnote of Table 5. (line 292-293)
Table 5. This reviewer understood how to read this table by the authors’ explanation. However, as it stands, this table misleads the reader. The followings are suggestions:
Comment 2. Titer should be outside parenthesis, but number should be in parenthesis, because this column indicates “Lung titer”.
RESPONSE: We have incorporated your comments by revising Table 5.
Comment 3. “ND” should be “-” instead of “not done”, and explain like “-“ stands for “virus not recovered” in a footnote.
RESPONSE: We replaced “ND” with “-” and explain as “Virus not recovered” in footnote (b). However, there is already “-” stand for “Data are not shown” in footnote (d), we have clarified by using different alphabet. (line 289, 291)
Comment 4. “b” should be attached to number and explain like “bNumber of animals in which the virus was detected by qRT-PCR per number inoculated”.
RESPONSE: Instead of attaching footnote “b” to number, we explained information you recommended sentence of footnote “a”. As Lung titer is located outside parenthesis and number located in parenthesis, we revised footnote “a”. (line 288-289).
Supplemental Figure S1 (a).
Comment 5. As this reviewer pointed out, “chicken” in an isolate name means that the virus was isolated from chicken, not human. Thus, vertical line with “Human” in this figure should not cover “chicken” virus even if their nucleotide identities are high.
RESPONSE: We have reflected this comment by revising supplemental Figure S1 (a). We think this change now better. We hope you agree.
Again, thank you for giving us the opportunity to strengthen our manuscript with your valuable comments and queries. We would be happy to make any further changes that may be required and hope that these revisions persuade you to accept our submission.
